# Experimental Study on the Mechanism of Enhanced Imbibition with Different Types of Surfactants in Low-Permeability Glutenite Reservoirs

**DOI:** 10.3390/molecules29245953

**Published:** 2024-12-17

**Authors:** Hongyan Qu, Jilong Shi, Mengyao Wu, Fujian Zhou, Jun Zhang, Yan Peng, Tianxi Yu, Zhejun Pan

**Affiliations:** 1National Key Laboratory of Petroleum Resources and Engineering, China University of Petroleum, Beijing 102249, China; shi2608857325@163.com (J.S.); wmy050626@163.com (M.W.); zhoufj@cup.edu.cn (F.Z.); 15166859902@163.com (J.Z.); 2College of Carbon Neutral Energy, China University of Petroleum, Beijing 102249, China; 3Unconventional Petroleum Research Institute, China University of Petroleum, Beijing 102249, China; 4College of Artificial Intelligence, China University of Petroleum, Beijing 102249, China; 5College of Petroleum, China University of Petroleum-Beijing at Karamay, Karamay 834000, China; yan.peng@cup.edu.cn; 6Engineering Technology Research Institute, PetroChina Xinjiang Oilfield Company, Karamay 834000, China; yutianxi2016@petrochina.com.cn; 7State Key Laboratory of Continental Shale Oil, Northeast Petroleum University, Daqing 163318, China; zhejun.pan@nepu.edu.cn

**Keywords:** enhanced oil recovery (EOR), low-permeability glutenite reservoirs, surfactant, imbibition, nuclear magnetic resonance (NMR)

## Abstract

Due to the complex physical properties of low-permeability glutenite reservoirs, the oil recovery rate with conventional development is low. Surfactants are effective additives for enhanced oil recovery (EOR) due to their good ability of wettability alteration and interfacial tension (IFT) reduction, but the reason why imbibition efficiencies vary with different types of surfactants and the mechanism of enhanced imbibition in the glutenite reservoirs is not clear. In this study, the imbibition efficiency and recovery of surfactants including the nonionic, anionic, and cationic surfactants as well as nanofluids were evaluated and compared with produced water (PW) using low-permeability glutenite core samples from the Lower Urho Formation in the Mahu oil field. Experiments of IFT, wettability, emulsification, and imbibition at high-temperature and high-pressure were conducted to reveal the underlying EOR mechanisms of different types of surfactants. The distribution and utilization of oil in different pores during the imbibition process were characterized by a combined method of mercury intrusion and nuclear magnetic resonance (NMR). The main controlling factors of surfactant-enhanced imbibition in glutenite reservoirs were clarified. The results demonstrate that the micropores and mesopores contribute most to imbibition recovery in low-permeability glutenite reservoirs. The anionic surfactant KPS exhibits a good capacity of reducing IFT, wettability alteration, and oil emulsification with the highest oil recovery of 49.02%, 8.49% higher than PW. The nonionic surfactant OP-10 performs well on oil emulsification and wetting modification with imbibition recovery of 48.11%. This study sheds light on the selection of suitable surfactants for enhanced imbibition in low-permeability glutenite reservoirs and improves the understanding of oil production through enhanced imbibition.

## 1. Introduction

The Mahu conglomerate reservoir is the largest super-large tight glutenite reservoir in the world with a reserve of 10 × 10^8^ t, which needs large-scale commercial development [1,2]. However, the reservoir is characterized by complex lithology, low porosity and ultra-low permeability, strong heterogeneity, poor reservoir properties, and low oil saturation [3,4,5], resulting in rapid production decline and low oil recovery, and diminishing EUR (Estimated Ultimate Recovery). Therefore, it is urgent to improve oil recovery of the conglomerate reservoir [6,7].

Large-scale volumetric fracturing with horizontal wells increases drainage area for enhanced oil recovery and provides large contact areas between fractures and matrix for imbibition [8,9]. The imbibition of hydraulic fracturing fluids plays a vital role in improving oil production in low-permeability and tight reservoirs, whereas the efficiency of spontaneous imbibition is relatively low [10,11,12,13,14,15,16]. The addition of surfactants can effectively improve the imbibition efficiency and oil utilization in the reservoir [17,18]. However, the mechanism of imbibition enhancement with different types of surfactants is complicated, especially for conglomerate reservoirs. In consequence, it is necessary to study the enhanced oil recovery mechanism of different surfactants and to optimize the injection parameters in low-permeability glutenite reservoirs.

The spontaneous imbibition in low-permeability reservoirs can be affected by various factors such as properties of rock, fluid, and environment [19,20,21,22,23], among which rock wettability and IFT are crucial. Surfactants can reduce the IFT between oil and water and the flow resistance of oil droplets in the pore throat [24,25]. In addition, the contact of surfactants with the rock surface can alter the rock wettability, improve the flow capacity of oil droplets, and increase the fluid imbibition distance [26,27]. Moreover, surfactants can emulsify oil, increase its dispersion in water, effectively reduce residual oil saturation, and improve oil recovery. However, the adsorption capacity of different surfactants varies in different types of reservoirs, considerably impacting their effects [28,29]. Therefore, the effects of different surfactants on imbibition need to be considered comprehensively to select appropriate surfactants.

Different surfactants are adopted due to distinctive reservoir properties. Common types of surfactants include cationic surfactants, anionic surfactants, nonionic surfactants, and amphoteric surfactants. Anionic and amphoteric surfactants perform better on shale rock than nonionic surfactants due to their good adsorption capacity on charged surfaces [30,31]. Since carbonate reservoirs are usually oil-wet or neutral-wet, surfactants with effects of wettability alteration need to be adopted to improve the hydrophilicity of carbonate reservoirs and enhance oil recovery. Compared with cationic surfactants and anionic surfactants, amphoteric surfactants can adsorb more on rock minerals and have better recovery potential [32,33].

Both anionic and nonionic surfactants can alter the wettability of sandstone reservoirs, whereas anionic surfactants exhibit better ability to increase the wettability of water than nonionic surfactants due to the increase in oil–rock negative charge, appropriately reducing the surface tension and improving the imbibition efficiency [34]. In addition, compound surfactants suit more types of reservoirs because of their excellent performance in reservoir adaptability, wetting alteration, reducing interfacial tension, and emulsifying oil [35,36,37]. Glutenite reservoirs mainly refer to the reservoirs composed of coarse clastic rock, consisting of sandstone particles with different particle size gradations [38,39]. They are characterized by low porosity, ultra-low permeability, and strong heterogeneity, resulting in low oil recovery efficiency. Therefore, technologies such as reservoir stimulation, imbibition with surfactants, and other measures need to be adopted to enhance recovery efficiency. However, the mechanism of imbibition of surfactants in low-permeability glutenite is rarely studied due to poor reservoir properties and the guidance for the surfactant selection is lacking.

In this study, the mechanism of enhanced imbibition of surfactants in low-permeability glutenite reservoirs was systematically investigated through a series of laboratory experiments. Core samples of a tight glutenite reservoir in Junggar Basin and different types of surfactants were selected. High-temperature and high-pressure imbibition experiments were carried out to quantify the oil utilization in different pores. In addition, the fluid distribution in the matrix pores was characterized by a combined method of mercury intrusion and NMR scanning. Moreover, the performance of IFT reduction, wettability alteration, and oil emulsification of different surfactants was evaluated according to the oil recovery. Furthermore, the mechanism of enhanced imbibition with different surfactants in low-permeability glutenite reservoirs was clarified. This study provides experimental foundations for the selection and design of surfactants for EOR in low-permeability glutenite reservoirs.

## 2. Results and Discussion

### 2.1. Effects of Surfactant Concentration on IFT 

The adhesions between oil and rock as well as oil and water decline when surfactants adsorb on the oil–rock and oil–water interfaces due to the reduction in IFT. Oil films are contracted and removed in the form of oil droplets under the action of buoyancy, and the flow resistance of oil droplets in the pore throat reduces [40,41]. However, a reasonable range of IFTs is needed to improve oil recovery because appropriate capillary pressure provides driving forces for imbibition. There is no regular relationship between oil recovery and IFT at present [34]. In this study, the IFT between oil and surfactant solutions with different concentrations (0.05%, 0.1%, and 0.15%) were measured, and the results are shown in Figure 1.

Compared with the PW, the addition of surfactants reduces the IFT, indicating that all these surfactants can decrease adhesion work and enhance gravity-driven imbibition. The anionic surfactants SDBS and KPS perform similarly the best on IFT reduction, and the IFT can be reduced to 0.62 mN/m and 0.58 mN/m at the concentration of 0.05% and further decreases with the increase in solution concentration. The cationic surfactant CTAB and anionic surfactant SDS perform similarly well with IFT under 1.5 mN/m at each solution concentration, while the IFT of SDS is higher than that of CTAB. In contrast, the IFT of the nonionic surfactant OP-10 remains the highest at 3.87 mN/m at the concentration of 0.05% and decreases to under 2.63 mN/m as the solution concentration increases to 0.15%. The IFT of the nanoemulsion CN-A is also relatively high at the concentration of 0.05%, whereas it decreases over 50% at the concentration of 0.1% and increases slightly as the solution concentration increases to 0.15%. It can be seen that surfactants SDBS and KPS have good advantages in reducing IFT, followed by CTAB and SDS. Surfactants CN-A and OP-10 do not perform as well as other surfactants while the IFTs are low enough compared to PW. In terms of enhanced oil recovery, there is no obvious relationship between IFT and imbibition recovery, and other factors need to be taken into account, which will be discussed in the following sections.

Compared with produced water (PW), the addition of surfactants can greatly reduce the IFT of the solution and enhance the imbibition. As the surfactant concentration increases, the IFT of most surfactant solutions decreases except for the nanoemulsion CN-A. The IFT of CN-A is the lowest at a concentration of 0.1%. For most surfactants, IFT can be reduced to similar values at the three concentrations, while for surfactants SDBS, CTAB, and OP-10, the decrease in IFT gradually slows down with the further increase in surfactant concentrations from 0.1% to 0.15%. Considering the cost of surfactants, the concentration of 0.1% was finally selected for the following experiments.

### 2.2. Wettability Alteration with Different Surfactants

The mineral compositions of all the core samples were obtained and analyzed through X-ray diffraction (XRD) before the measurements of contact angle, as shown in Table 1.

Quartz and plagioclase accounts for 31.6% and 29.7% of the mineral composition in the rock, respectively. According to previous studies, quartz is composed of Si-O tetrahedra, and covalent electrons are biased towards silicon atoms. There are siloxane functional groups (Si-O-Si) on the surface, which react with water to produce surface hydroxylation and enhance the hydrophilicity of the quartz surface [42,43]. Plagioclase is a silicate mineral, and its crystal structure is composed of a three-dimensional structure of two connected silicon-oxygen and aluminum-oxygen tetrahedra. The isomorphic substitution (Si^4+^ is substituted by Al^3+^ in the silicon-oxygen tetrahedra) results in the negative charge of the lamellar structure, and the cations can adsorb between the layers [44]. Since the content of hydrophilic minerals in the rock is high, water in the formation is easily adsorbed on the surface of the rock, exhibiting hydrophilic characteristics [45].

The contact angle of rock slices changes after imbibition in different surfactant solutions, as shown in Figure 2. The core samples saturated with oil were initially weak water-wet or intermediate-wet with contact angles ranging from 50° to 70°. The initial contact angle with all the surfactant solutions is relatively low, indicating that the wettability of the glutenite core surface cannot be altered completely to oil-wet by oil aging.

After soaking with different surfactant solutions for 2 days, the wettability changed to stronger water-wet in SDS and SDBS solutions with the contact angle decreasing by 17.30° and 16.25°, respectively. In particular, SDS increased water wetness the most. The contact angle reduced by 11.4° and 10.9°, respectively, for OP-10 and KPS, while little change was observed for that of CTAB and CN-A. Consequently, the capillary pressure acts as the driving force in the SDS, SDBS, KPS, and OP-10 solutions, and oil recovery is improved. The main mechanism of EOR of the anionic and nonionic surfactants is the reduction in the contact angle. Similarly, the contact angle reduces as well with the oil displacement agent used in the field, which is one of the main mechanisms of EOR improvement. Therefore, the potential of surfactants for EOR is related to the capability of wettability alteration. The ion-pair mechanism is assumed to be the main reason for the wettability alteration of cationic surfactants [46]. However, the contact angle of CTAB solutions does not change much in this study, demonstrating that the ion-pair mechanism of cationic surfactants may not be the dominant factor in wettability alteration and imbibition efficiency improvement in glutenite reservoirs.

### 2.3. Emulsification Performance

The emulsification performance of different surfactant solutions were recorded at 30 s, 60 s, and 600 s, respectively, after heating and fully shaking, as shown in Figure 3. The anionic surfactants SDBS and KPS perform the best, and the demulsification rate for static 30 s is almost 0, followed by SDS and OP-10. The emulsification performance of PW, CN-A, and CTAB is relatively poor because the demulsification rate is approximately 100% within 30 s after shaking.

The oil emulsification performance is one of the main mechanisms of EOR for KPS and OP-10. Low IFT facilitates the emulsion, but it is not conducive to the stability of the emulsion [47]. The effect of OP-10 on emulsifying oil may be attributed to its high IFT, which needs to be further studied. In summary, there is a certain relationship between demulsification performance and imbibition recovery, but there is no good correlation in this study.

### 2.4. Pore Distribution and Classification of Glutenite Cores

MIP measurements were conducted to obtain the cumulative distribution of the pore radius, while the cumulative distribution of the relaxation time was obtained from the *T_2_* spectrum with NMR. The cumulative distribution curves of NMR *T_2_* relaxation time and pore radius of high-pressure MIP were drawn in the same logarithmic coordinate diagram, as shown in Figure 4a. Since the MIP cannot reflect the pore size distribution over the maximum mercury pressure, only some of the points on the cumulative distribution curve of the relaxation time were selected to correspond to the pore throat converted through the pore distribution of MIP.

Each cumulative distribution frequency corresponds to a specific group of pore radius and a certain group of relaxation time. For the same cumulative distribution frequency, the corresponding pore radius and *T_2_* relaxation time can be selected and plotted as the y and x axes, as shown in Figure 4b. The relationship between them could be derived by fitting the curve with the linear least square method, and the constants C and n in Equation (10) can be determined, respectively.

The quantitative relationship between the pore radius and *T_2_* relaxation time can be expressed as follows:(1)$$r_{t}=0.013T_{2}^{0.7263}$$

The relationship was used to classify the pores in glutenite cores and study the oil recovery during the imbibition experiments. According to the range of pore radius, the pore space can be divided into three types, including micropores (0–0.01 μm), mesopores (0.01–0.1 μm), and macropores (>0.1 μm), as shown in Table 2 and Figure 5. The *T_2_* spectrum exhibits a trimodal distribution with three distinct peaks. The pore size distribution obtained by MIP is similar to the *T_2_* spectrum, as shown in Figure 5. The good consistency in shape and amplitude of the two curves demonstrates the rationality of the curve fitting method.

The volume fractions of micropores, mesopores, and macropores were calculated according to the pore classification criteria shown in Table 2. The amplitude of NMR signals of oil-saturated core samples is illustrated in Figure 6a, and the proportion of different pore sizes of each core was calculated accordingly, as shown in Figure 6b. The pores of seven core samples are mainly dominated by micropores, accounting for 52.6% to 67.5% of the total pore volume. Mesopores account for 29.1~35.6% while the proportion of macropores is relatively low, only 2.34~15.4%. Therefore, most of the saturated oil is stored in the micropores.

### 2.5. NMR-Monitored HTHP Imbibition with Different Surfactants

#### 2.5.1. HTHP Imbibition

The experiments of high-temperature and high-pressure imbibition were conducted at 75 °C and 24 MPa, and monitored with NMR at 0 h, 1 h, 3 h, 6 h, 12 h, 20 h, 30 h, 48 h, 72 h, and 120 h during the imbibition. Core samples from the low-permeability conglomerate reservoir with surfactants at the concentration of 0.1% and pure PW were used to study the effect of different surfactants on the imbibition efficiency and oil recovery.

The *T_2_* spectrum was recorded during the HTHP imbibition with different surfactant solutions, as shown in Figure 7. The NMR amplitude continues to decrease as the imbibition proceeds, indicating that the aqueous solutions invade the rock pores and the oil inside is displaced. The oil saturation decreases and the water saturation increases mostly near the fracture and core surface. The evolutions of the NMR signals are similar for all the surfactant solutions. Specifically, the magnitudes decrease dramatically in the micropore (short relaxation time) while those in the mesopore and macropores (long relaxation time) change slightly. This demonstrates that oil was mainly distributed in the micropores initially and displaced to the mesopores and macropores by the surfactant solutions and PW and extracted from the fractures during the imbibition. Therefore, the oil recovery is attributed to the utilization of oil in micropores to a large extent.

The performance on the oil displacement and oil recovery of different solutions varies. The *T_2_* spectrum decreases more during the imbibition with the surfactant solutions compared to the PW. Therefore, the addition of surfactants contributes to imbibition, oil displacement, and enhanced oil recovery in the glutenite reservoirs. The macropores in the core contribute more to oil recovery with PW, while surfactants can facilitate the oil utilization in the micropore and the NMR signal reduces more significantly in each interval of the imbibition. However, there are obvious differences in the efficiency and rate of imbibition between different types of surfactants.

There is little oil left in the core with KPS and OP-10 solutions after imbibition, as shown in the core image in Figure 7. As mentioned above, the main factors affecting the imbibition recovery should be the reduction in IFT and wettability alteration.

#### 2.5.2. The Imbibition Efficiency of Different Surfactants

The imbibition efficiencies of the six surfactant solutions and PW were investigated in this section. The imbibition of all the solutions mainly occurs within 20 h, as shown in Figure 8a. The anionic surfactant KPS achieved the highest imbibition recovery, closely followed by the nonionic surfactant OP-10, which are higher than those of the cationic CTAB and anionic SDS and SDBS, indicating the high EOR potentials of anionic and nonionic surfactants. The imbibition efficiency of the nanofluid CN-A is the lowest among all the surfactant solutions, slightly higher than that of PW, indicating poor applicability of the nanoemulsion in this conglomerate reservoir.

According to the evolution of imbibition recovery with time, as shown in Figure 8a, the imbibition rate was calculated accordingly, as shown in Figure 8b, which is consistent with the recovery. There is a peak imbibition rate at the very beginning of the imbibition in each case, indicating rapid imbibition on the surface of the core and fractures in the first 1~3 h. The peak rate is similarly the highest for KPS and OP-10, which reaches 0.22%/min after 1 h imbibition but decreases rapidly to about 0.03%/min at 3 h, followed by a gradual decrease until it levels off after 20 h. The other surfactant solutions exhibit similar evolution trends, except that the imbibition rate slightly increases from 3 to 6 h. Interestingly, the imbibition rate of CTAB is the lowest in the first hour, less than that of the PW.

The electrostatic interaction among oil, rock, and surfactant affects the wettability alteration [48]. The negative charges of both oil and rock result in weak adhesion strength between oil and rock. It is easy for oil to desorb from the rock surface due to the electrostatic repulsion, leading to the intermediate-wet rock surface. The adsorption of anionic and nonionic surfactants on the rock surface can reduce the zeta potential on the rock surface and enhance the electronegativity of the rock surface. Anionic surfactants increase the negative charge of the rock surface due to their negative heads, and nonionic surfactants increase the electronegativity of the rock surface due to the presence of a large number of hydroxyl groups in their molecular structure [34]. Specifically, the anionic surfactants KPS and SDBS as well as the nonionic surfactant OP-10 increase the negative charges of the rock surface, enhancing the electrostatic repulsion between oil and rock, intensifying the oil strip from the rock surface. In contrast, the electrostatic attraction is higher for the positively charged CTAB [34], which can adsorb on the rock more easily. In consequence, oil tends to adhere on the surface because of the reduced electrostatic repulsion and the imbibition rate of CTAB is the lowest. Therefore, the anionic and nonionic surfactants are more suitable for the conglomerate reservoir in terms of EOR while the cationic surfactant performs relatively poorly.

#### 2.5.3. The Oil Recovery in Different Pores

The final imbibition recovery of different surfactants in 5 days and the imbibition recovery in different pores was calculated and compared according to the relationship between *T_2_* and pore throat radius, as shown in Figure 9a and Figure 9b, respectively.

The ultimate imbibition recovery of all surfactant solutions is higher than that of pure PW after 120 h of imbibition. However, the ultimate recovery of oil with different surfactants is quite different. Some surfactant solutions perform better, while the effects of others are not obvious. The order of different surfactants in terms of ultimate recovery is KPS > OP10 > SDBS > CTAB > SDS > CN-A > PW. KPS and OP-10 with good salt resistance are more suitable for this reservoir condition. KPS and OP-10 have better performance on imbibition enhancement and oil recovery than other surfactants. The imbibition recovery of the surfactant KPS is the highest, reaching 49.02%, 7.37% and 8.49% higher than that of CN-A and PW, respectively. The imbibition recovery of the nonionic surfactant OP-10 is slightly lower than that of KPS, which is 48.11%, 6.46% and 7.58% higher than that of CN-A and PW, respectively.

According to the relationship between the *T_2_* relaxation time and pore throat radius as shown in Equation (1), and the pore classification standard as shown in Table 2, the changes in amplitude of oil recovery in different pores can be calculated as follows:(2)$$R_{i}=\frac{S_{i1}-S_{i2}}{S_{t}}\times 100\%$$
where $R_{i}$ is the contribution to recovery efficiency of certain pores, %; $S_{i1}$ is the envelope area of the *T_2_* spectrum in the initial saturated oil state in certain pores; $S_{i2}$ is the envelope area of the *T_2_* spectrum after imbibition in certain pores; $S_{t}$ is the envelope area of the *T_2_* spectrum in the initial saturated oil state in all the pores of the core; i = 1, 2, 3, representing micropores, mesopores, and macropores, respectively.

Since the volume fractions of mesopores and macropores are much smaller compared to those of micropores, the oil recovery in these pores is relatively lower. About 29.57% of the oil in the micropores and 10.40% in the macropores was recovered when PW was used as the imbibition solution. The addition of surfactants cannot enhance the oil recovery in the macropores much but further improves the oil recovery in the micropores and mesopores, from 31.86% to 35.33% and 3.25% to 7.88%, respectively.

The nonionic surfactant OP-10 performs the best on improving oil recovery in micropores by 35.33%, followed by the anionic surfactant KPS. The oil recovery of SDBS and CTAB in micropores is 34.33% and 34.26%, respectively, which is slightly lower than those of KPS and OP-10. The performance of CN-A is the poorest, which increases the oil recovery in the micropores by only 2.29% compared to PW. Therefore, for low-permeability glutenite reservoirs, the key to improving imbibition recovery is the utilization of oil in micropores and mesopores. The anionic surfactant KPS and nonionic surfactant OP-10 have better effects on imbibition in micropores and mesopores than PW and other surfactants.

When PW is used as the imbibition solution, oil migration is mainly driven by capillary force in pores, where water-wet conditions are necessary. As mentioned above, the minerals in the glutenite rock are mainly quartz and feldspar, which are initially or easy to be changed to water-wet. Though the oil recovery improvements in the micropores are significant in this study, the addition of surfactants can enhance the oil recovery from the mesopores and macropores as well. The oil recovery in the mesopores is only 0.56% with PW, which is improved slightly by KPS to 3.25%, and further to 5.43%~7.88% using other surfactants. The cationic surfactant CTAB performs the best on the oil recovery in mesopores. The oil recovery of PW in macropores is the highest, as much as 10.51%, followed by KPS, while the recovery of other surfactants in macropores is 3.56%~5.56%. In general, the total oil recovery of KPS in both mesopores and macropores remains the highest at 13.76% among all surfactants, followed by 12.78% of OP-10. Therefore, the anionic surfactant KPS and the nonionic surfactant OP-10 perform well on the oil recovery improvement in both small pores and large pores.

#### 2.5.4. The Mechanism of Enhanced Imbibition with Different Surfactants

As studied above, the mechanism of enhanced imbibition with surfactants is related to IFT reduction, wettability alteration, and oil emulsification, as shown in Figure 10. The IFT of most surfactant solutions is inversely proportional to the total oil recovery. The adsorption of surfactant molecules reduces the interfacial energy of the system, leading to a decrease in IFT. However, there is no definite relationship between the IFT and oil recovery [34]. The oil recovery of anionic surfactant KPS is the highest, whereas the IFT is not the lowest because appropriate capillary pressure is needed for the driving force in the imbibition. The relatively low IFT with KPS is suitable for oil recovery improvement. The moderate decrease in surface tension reduces the adhesion between the oil and rock surface, which leads to the detachment of oil droplets, reduces the saturation of remaining oil, and improves oil recovery. Therefore, the appropriate IFT is needed for a relatively high oil recovery.

The oil recovery of the nonionic surfactant OP-10 is second to KPS, whereas the IFT is relatively high, indicating that wettability alteration may be the main mechanism of enhanced oil recovery. The oil and water distribution in the rock is controlled by wettability, and the capillary force increases with water wetness as the driving force. The anionic surfactants SDS, SDBS, and KPS and the nonionic surfactant OP-10 perform well on altering rock wettability and oil emulsification, increasing the capillary force, enhancing the imbibition to displace oil, and improving oil recovery. Surfactants enhance imbibition and oil recovery by reducing IFT, altering wettability, and emulsifying crude oil. Moderate reduction in IFT reduces the adhesion of oil to the rock surface, facilitating the falling off of oil droplets from the rock surface, whereas certain capillary force is still required for driving oil in oil recovery. In addition, the wettability can be altered to be more hydrophilic in favor of oil recovery and oil emulsification, which improves oil migration in the pore throat with surfactants. However, the ability to reduce IFT, change wettability, and emulsify oil are different for different surfactants due to their different composition, structure, and mechanism of enhancing imbibition. Therefore, surfactants perform differently in enhancing imbibition and oil recovery. It is necessary to consider all the factors for selecting surfactants to improve the oil recovery in the glutenite reservoirs.

In summary, the ability to utilize oil in micropores and mesopores is the key to enhancing oil recovery. The final oil recovery rate is closely related to the properties of the surfactants. Specifically, the anionic surfactant KPS with the highest imbibition recovery rate has a good ability to reduce IFT, wettability alteration, and oil emulsification in low-permeability glutenite reservoirs. The nonionic surfactant OP-10 has good effects on oil emulsification and certain effects on wetting modification of reservoir rocks. The anionic surfactants SDBS and SDS play good roles in IFT reduction, oil emulsification, and wetting modification. In summary, various factors need to be considered for the optimization of the surfactants suitable for glutenite reservoirs. According to the final oil recovery rate, the anionic surfactant KPS may be a good choice for these low-permeability glutenite reservoirs.

## 3. Materials and Methodology

### 3.1. Experimental Materials

#### 3.1.1. Preparation and Properties of Fluids

Since gases are released from the oil after oil is produced from the formation, the density and viscosity of the oil increase. To keep the density and viscosity consistent with the oil in formation conditions, oil produced from a tight glutenite reservoir in the Junggar Basin was diluted with kerosene with a volume ratio of 1:3, as shown in Figure 11a. The density and viscosity of the synthetic oil were 0.88 g/cm^3^ and 1.15 mPa·s, respectively, at 75 °C and 23.94 MPa, similar to the reservoir conditions.

Water used in the experiments was obtained from the produced water (PW) in the same reservoir, as shown in Figure 11b. The water type was NaHCO_3_, and the ion composition is shown in Table 3. The detailed properties of synthetic oil and PW at 75 °C are shown in Table 4. Both the synthetic oil and the PW were filtered with membranes to remove impurities before the experiments.

#### 3.1.2. Preparation of Surfactants and Solutions

Reservoirs with high temperature and high salinity present high demands of temperature and salt resistance for surfactants. In this study, three types of commercial surfactants were selected, including nonionic sulfonate surfactants (SDS and SDBS), a nonionic surfactant polyoxyethylene ether (OP-10), and a cationic surfactant (CTAB). In addition, a nanoemulsion (CN-A) was self-developed and an oil displacement agent (KPS) was obtained from the Xinjiang oilfield. The detailed descriptions of each surfactant are presented in Table 5.

High concentrations of surfactant solutions were prepared with the produced water at first and then diluted to the target concentrations of 0.05%, 0.1%, and 0.15%. In addition, 15% Mn^2+^ solvent was added into the surfactant solutions to shield the hydrogen proton signal in the solution during the NMR-monitored imbibition. All solutions were stirred thoroughly for a full dissolution. The Krafft temperatures (KPs) of the ionic surfactants SDS, SDBS, and CTAB were 13–18 °C, 27.6 °C, and 24–27 °C, respectively [49,50], and the cloud point (CP) of the nonionic surfactant OP-10 was 61–67 °C. Since the experimental temperature conditions are much higher than the KPs of the three ionic surfactants, they could be completely dissolved in PW at the experimental temperature and this ensures the stability of these surfactant solutions. For the nonionic surfactant OP-10, the experimental temperature is slightly higher than its turbidity point, but according to the results of this study, the stability of OP-10 is good and has a higher effect on improving oil recovery.

#### 3.1.3. Preparation of Core Samples

Four full-diameter core samples were drilled and obtained from a tight glutenite reservoir in the Junggar Basin at depths from 2787 m to 2791 m, which were precisely cut into ϕ 25 mm (in diameter) × 50 mm (in length) cylinders and ϕ 25 mm (in diameter) × 2 mm (in length) slices, as shown in Figure 12a,b. Core samples with similar rock properties were selected for comparative experiments. Seven cylindrical and seven sliced core samples were selected for experiments of high-temperature and high-pressure imbibition and wettability evaluation, respectively. In addition, some particles were prepared for the mineral composition analysis through X-ray diffraction (XRD).

Porosity and permeability of the cylindrical core samples were measured with the helium expansion and pulse decay methods, respectively, after the core samples were cleaned and dried in an oven at the temperature of 105 °C. The porosity is approximately 7.55% on average and permeability ranges from 0.0015 mD to 0.0091 mD. The detailed petrophysical properties of the core samples are listed in Table 6. The cylindrical core samples were split off by the Brazilian splitting method to simulate high-temperature and high-pressure imbibition from fractures to the matrix because of the low initial matrix permeability.

The dried cylindrical core samples were evacuated for 12 h to remove the air in the cores, and saturated and aged in the oil tank with synthetic oil at a reservoir temperature of 75 °C and reservoir pressure of 25 MPa for 20 days before the imbibition experiments. The sliced core samples were evacuated and saturated with oil for over 72 h to simulate the reservoir wettability condition.

### 3.2. Experimental Methodology

In this study, the fluid inside different pores was characterized through high-temperature and high-pressure imbibition experiments, combining the pore characterization methods of mercury intrusion and NMR *T_2_* spectrum. In addition, the oil utilization in each imbibition stage with different types of surfactants in the low-permeability glutenite reservoirs was revealed. Moreover, the mechanism of enhanced oil recovery in the glutenite reservoirs was clarified through a series of experiments including interfacial tension, wettability change, and oil emulsification performance with different surfactants. Furthermore, suitable surfactants were selected for the reservoir conditions combining the results of imbibition recovery.

#### 3.2.1. Oil–Water Interfacial Tension

The IFT between each surfactant solution and synthetic oil was measured by the hanging ring method (du Noüy method) with the BZY-2 micro-controlled automatic surface and interface tensiometer (Shanghai Hengping Instrument Factory, Shanghai, China), as shown in Figure 13a,b. The IFT of each surfactant was tested at three concentrations of 0.05%, 0.1%, and 0.15%, and the effect of surfactant concentration on the IFT was investigated. In this method, the surfactant solution is poured into the glass vessel (about 10 mm high), which is then placed in the middle of the tray. The tray is raised until the platinum ring immerses 5–7 mm deep in the solution. The synthetic oil is poured above the surfactant solution to keep the oil–water interface for 30 s. When the platinum ring is removed from the oil–water interface, the maximum tension required is equal to the sum of the weight of the ring itself, the product of the interfacial tension, and the circumference of the oil–water interface. The measurements were conducted three times for each surfactant and PW and the value of IFT was averaged to ensure the reliability of the obtained IFT results.

#### 3.2.2. Wettability Evaluation by Contact Angle

The contact angle of the water (surfactant solution)/oil (synthetic oil)/rock (glutenite) system was measured by the captive bubble method with a JY-PHb contact angle measuring instrument (Chengde Jinhe Instrument Manufacturing Co., LTD, Chengde, China), as shown in Figure 14a. The contact angle was measured in comparative groups including PW with 0.1% different surfactants and PW without surfactants.

The contact angle of each rock sample was measured before and after it was soaked in each surfactant solution for 72 h, respectively. The ability of wettability alteration of different surfactants was evaluated according to the change in contact angle. In this method, the sliced sample is placed on the holder in the measurement chamber, and the surfactant solution is gradually injected into the chamber until the subsurface of the sample is submerged in the solution. An oil droplet is injected using an inverted needle and attached to the subsurface of the sample. After stabilization, the image of the oil droplet is taken by a high-resolution microscopic camera of the CA instrument, and the static contact angle is measured, as shown in Figure 14. Each sample was measured 3–5 times and the average value was calculated to ensure the consistency of the measurements.

#### 3.2.3. Emulsification

Surfactants adsorb on the water–oil interface and form a water–oil mixed interface film according to the hydrophilicity and hydrophobicity of the surfactant molecules. The IFT changes due to the emulsification, reducing the oil viscosity and the flow resistance between oil droplets and rock surface, and improving the oil fluidity [51,52]. In the complex reservoir environment, high stability for the emulsion is required to reduce demulsification and improve oil recovery [53].

In this study, the emulsifying capability of each surfactant was determined. The surfactant solutions at concentration of 0.1% were added to 50 mL scale tube with the synthetic oil at a volume ratio of 7:3. The scale tube was placed in a water bath and heated for 20 min at a temperature of 75 °C. The test tube was taken out for full oscillation, and the oil precipitation was compared at 30, 60, and 600 s.

#### 3.2.4. NMR-Monitored High-Temperature and High-Pressure Imbibition

Spontaneous imbibition was commonly adopted for monitoring oil recovery using the volume method with an Amott cell [54,55,56,57,58]. However, the volume of oil recovered from the low-permeability and tight rock is too little and cannot be monitored precisely. In addition, there are pressure differences between the fracture and matrix at the beginning of the well shut-in. Therefore, high-temperature and high-pressure (HTHP) imbibition experiments were carried out in this study. The effects of different types of surfactants on oil recovery were compared during the HTHP imbibition experiments with core samples with similar petrophysical properties. The migration of oil in the rock pores with different surfactants was characterized by the changes in the *T_2_* spectrum obtained with a low-field MacroMR12-150H-I nuclear magnetic resonance (NMR) instrument (Suzhou Newmai Corporation, Suzhou, China), as shown in Figure 15.

Since 15% Mn^2+^ solvent was added into the surfactant solutions to shield the hydrogen proton signal during the NMR-monitored imbibition, only the oil phase within the rock shows the *T_2_* spectrum signals under the magnetic field condition. Before the imbibition, the cylindrical core samples were dried and saturated with oil, and the initial *T_2_* spectra of the core samples were measured after oil aging. The imbibition experiments were conducted in the imbibition tank under high-temperature and high-pressure conditions, as shown in Figure 15, using different surfactant solutions at a concentration of 0.1% and the PW as comparative groups. The tank was heated and maintained at 75 °C, and a constant pressure of 24 MPa was set. The *T_2_* spectrum was recorded and analyzed at selected time intervals (1 h, 3 h, 6 h, 12 h, 20 h, 30 h, 48 h, 72 h, 120 h), and the evolution of oil recovery rate with time and the ultimate recovery were calculated.

#### 3.2.5. Pore Size Characterization

The number of hydrogen atoms presenting in a fluid in a porous medium can be detected through the transverse relaxation time (*T_2_*) according to the NMR theory. Therefore, the total pore size and pore size distribution can be characterized with single-fluid saturated cores. The total *T_2_* relaxation time can be expressed as follows [59]:(3)$$\frac{1}{T_{2}}=\frac{1}{T_{2,bulk}}+\frac{1}{T_{2,surface}}+\frac{1}{T_{2,diffusion}}$$
where *T_2_* is the transverse relaxation time, ms; $T_{2,bulk}$ is the bulk relaxation time, ms; $T_{2,surface}$ is the surface relaxation time, ms; $T_{2,diffusion}$ is the diffusion-induced relaxation time, ms.

Since the overall relaxation time is usually significantly longer than the surface relaxation time, the bulk relaxation time can be ignored. In addition, the diffusion relaxation time can also be ignored in a uniform magnetic field. The transverse relaxation time can then be simplified [60]:(4)$$\frac{1}{T_{2}}=\rho_{2}\frac{S}{v}$$
where $\rho_{2}$ is the transverse surface relaxation intensity, which depends on the pore surface properties, mineral composition, and saturated fluid properties, μm/ms; and S/v is the specific surface area of a single pore, μm^2^/μm^3^.

The specific surface area of a single pore is a function of pore radius and pore shape factor, which can be described as [61]:(5)$$\frac{S}{v}=\frac{F_{s}}{r}$$
where r is the pore radius, mm; and $F_{s}$ is the pore shape factor, *F_s_* = 2 for cylindrical pores and *F_s_* = 3 for spherical pores.

By substituting Equation (5) into Equation (4), the relationship between the pore size and *T_2_* spectrum can be obtained [62,63,64]:
(6)$$r=T_{2}\rho_{2}F_{s}$$

The relationship between the pore radius and *T_2_* spectrum can also be power exponential because of the complex pore structure in low-permeability or tight glutenite as follows [65,66]:(7)$$r^{n}=T_{2}\rho_{2}F_{s}$$

Since *ρ_2_* and *F_s_* are constants for a fluid-saturated core, the coefficient *C* can be introduced and defined as:(8)$$C={\rho_{2}F}_{s}$$

Substituting Equation (8) into Equations (6) and (7), the relationship between pore radius and *T_2_* spectrum can be simplified:(9)$$r^{n}={CT}_{2}$$

The pore radius can be calculated as follows:(10)$$r=C^{\frac{1}{n}}{T_{2}}^{\frac{1}{n}}$$

However, the actual size of the pore space cannot be determined by NMR alone. In practical applications, the pore structure needs to be characterized by combining the NMR *T_2_* spectrum with other methods such as high-pressure mercury intrusion porosimetry (MIP) to describe the entire scale of the pore size in real cores.

According to the capillary pressure, the distribution of the pore throat radius determined by MIP can be described as follows:(11)$$r_{t}=\frac{2\sigma \cos\theta }{p_{c}}$$
where *r_t_* is the pore throat radius, μm; θ is the contact angle; σ is the interfacial tension, N/m; and $p_{c}$ is the capillary pressure, MPa;

From previous studies, the pore size distribution can be characterized by combining the MIP and cumulative pore size distribution of the *T_2_* spectrum. Therefore, the relationship between pore radius r and transverse relaxation time $T_{2}$ can be obtained with the fitted parameters C and n.

## 4. Conclusions

In this study, the effects of different surfactants on the imbibition recovery enhancement in low-permeability glutenite reservoirs were evaluated. The mechanism of enhanced imbibition was studied based on a series of surfactant-related performance tests. The applicability of different surfactants in low-permeability glutenite reservoirs was analyzed. Some conclusions were drawn as follows:

(1) The glutenite rocks are mainly composed of quartz and feldspar, which are initially or easy to change to water-wet. The anionic surfactants SDS, SDBS, KPS, and nonionic surfactant OP-10 perform well on rock wettability alteration;

(2) The key factor to improve imbibition recovery in glutenite reservoirs is to utilize the oil in the micropores. The micropores and mesopores account for 52.6%~67.5% and 29.1%~35.6% of the total pore volume, respectively. The addition of surfactant significantly improves the oil recovery in micropores and mesopores;

(3) The anionic surfactant KPS and nonionic surfactant OP-10 improves oil recovery by 8.49% and 7.58%, respectively, compared to PW, showing good ability on IFT reduction, wettability alteration, and oil emulsification, which are the main mechanisms of enhanced imbibition recovery in low-permeability glutenite reservoirs.

## Figures and Tables

**Figure 1 molecules-29-05953-f001:**
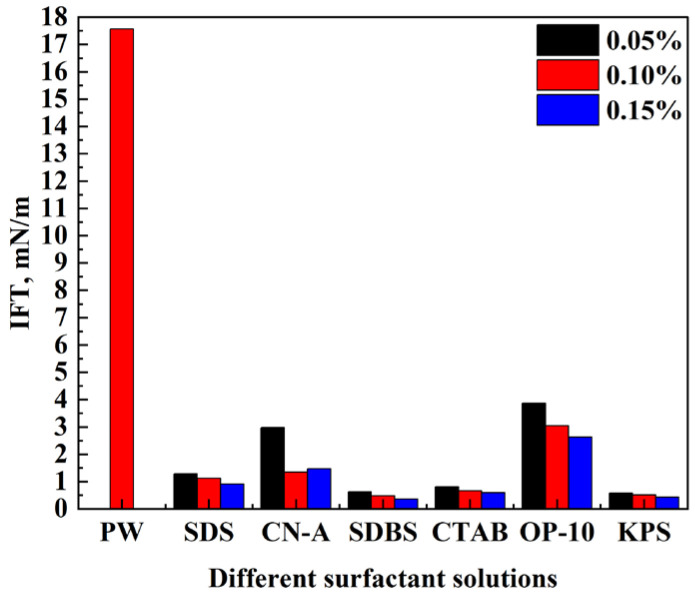
Oil–water IFT of different surfactant solutions with different concentrations.

**Figure 2 molecules-29-05953-f002:**
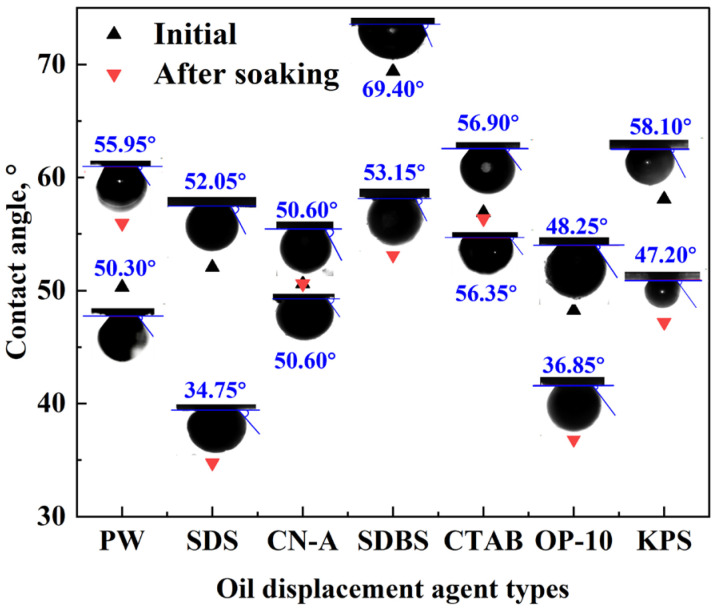
The change in contact angle with the surfactant solution before and after soaking.

**Figure 3 molecules-29-05953-f003:**
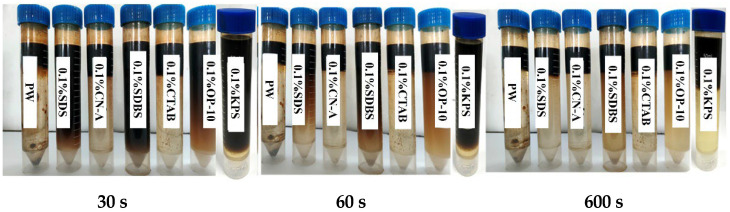
Emulsification effects of different surfactants at different standing times.

**Figure 4 molecules-29-05953-f004:**
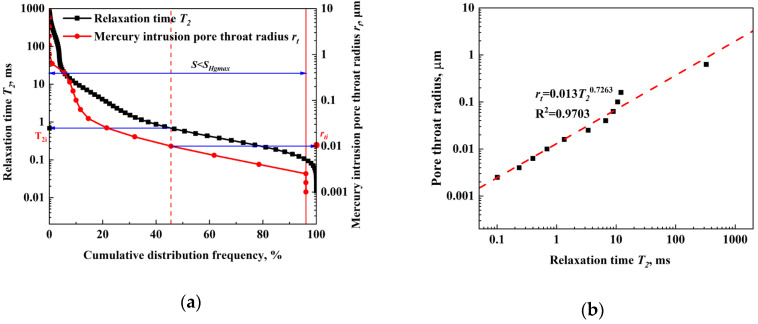
Conversion of pore radius and relaxation time: (**a**) cumulative distribution curves of *T_2_* relaxation time from NMR and pore radius from MIP with frequency; (**b**) fitting curves.

**Figure 5 molecules-29-05953-f005:**
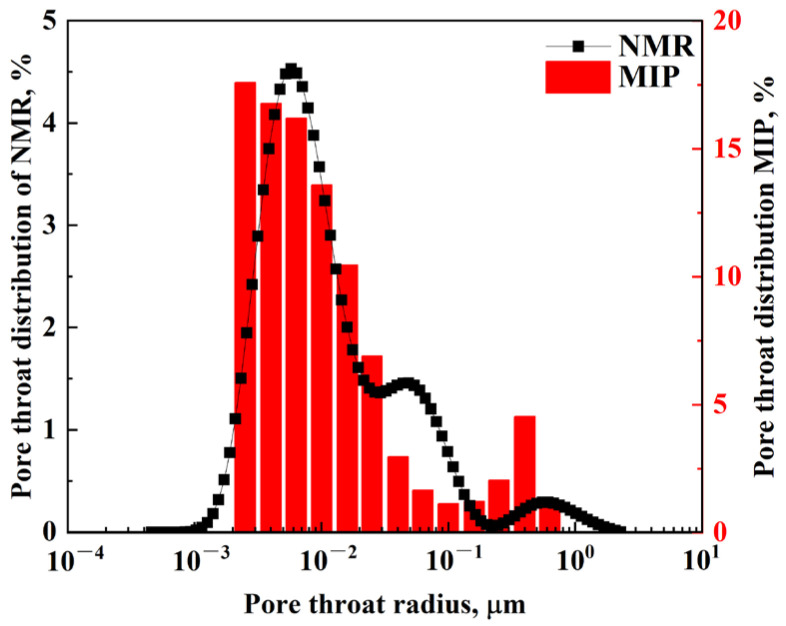
Pore size distribution of glutenite cores based on NMR and high-pressure MIP.

**Figure 6 molecules-29-05953-f006:**
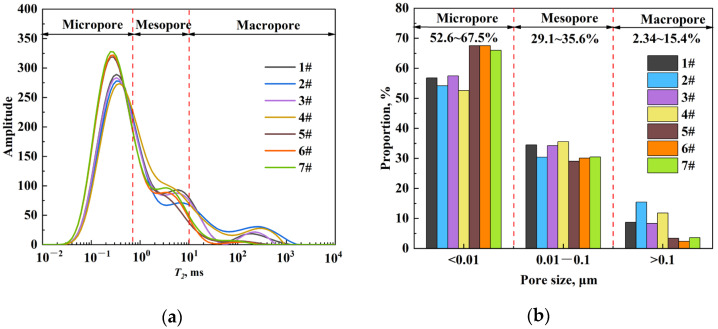
Classification of pores of all the core samples: (**a**) *T_2_* relaxation distribution and (**b**) the volume fractions of micropores, mesopores, and macropores in the samples.

**Figure 7 molecules-29-05953-f007:**
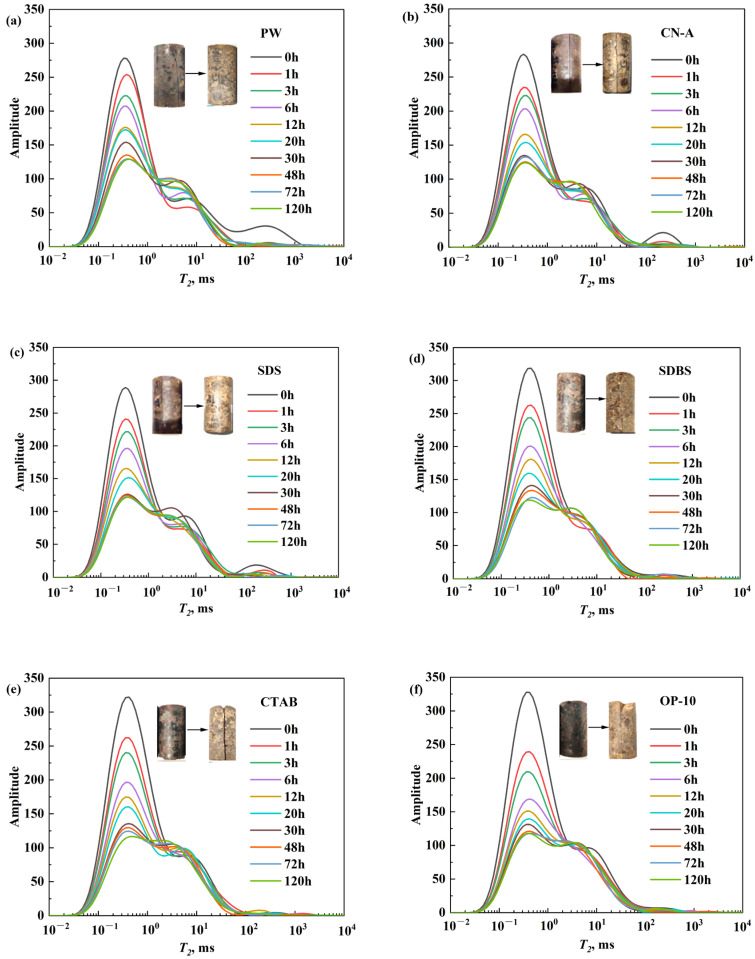
Evolution of *T_2_* spectra during imbibition with different surfactant solutions and PW: (**a**) PW; (**b**) CN-A; (**c**) SDS; (**d**) SDBS; (**e**) CTAB; (**f**) OP-10; (**g**) KPS.

**Figure 8 molecules-29-05953-f008:**
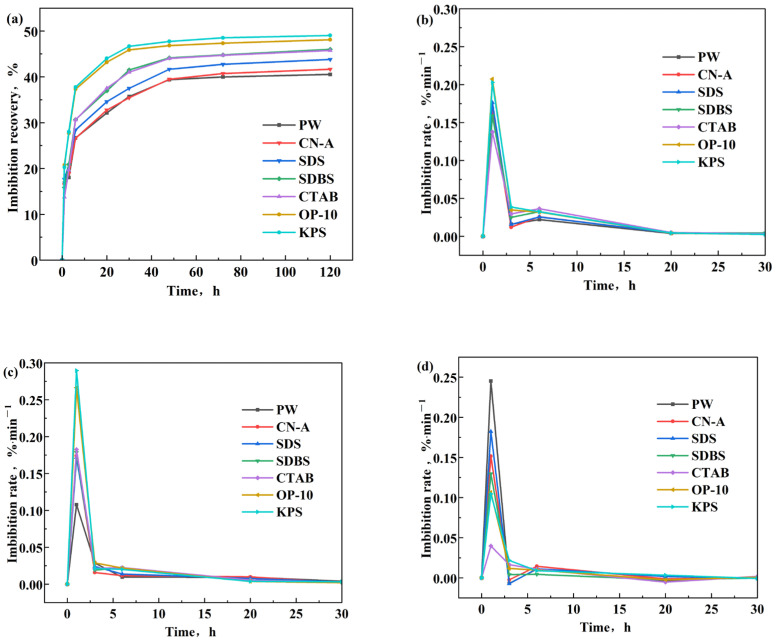
Changes of imbibition recovery and imbibition recovery rate of different surfactant solutions: (**a**) imbibition recovery; (**b**) imbibition recovery rate in all pores; (**c**) imbibition recovery rate in micropores; (**d**) imbibition recovery rate in mesopores and macropores.

**Figure 9 molecules-29-05953-f009:**
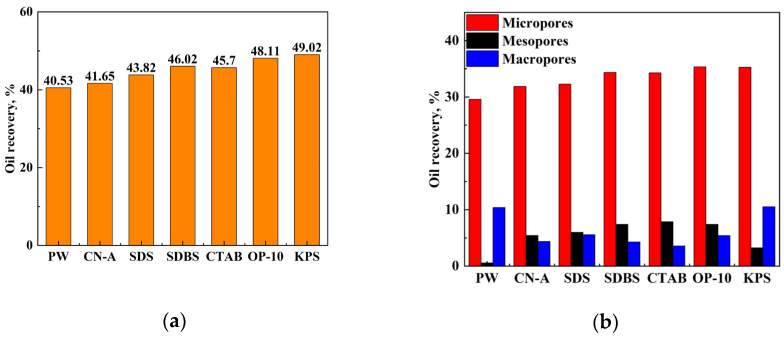
Total imbibition recovery of different surfactant solutions: (**a**) imbibition recovery; (**b**) contribution of different pores to oil recovery with different surfactants.

**Figure 10 molecules-29-05953-f010:**
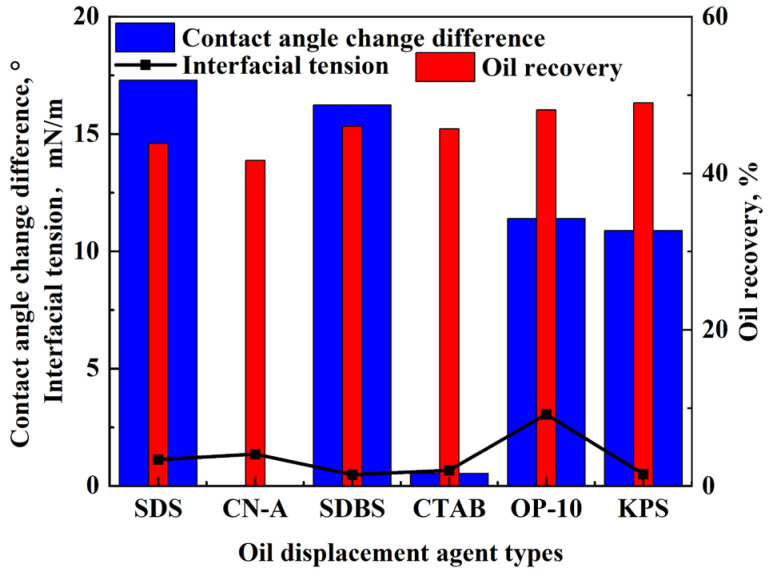
The relationship between oil recovery and surfactant properties.

**Figure 11 molecules-29-05953-f011:**
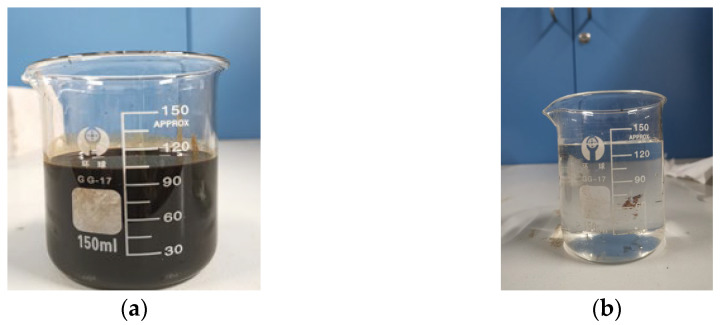
Synthetic oil and PW for the experiments. (**a**) Synthetic oil. (**b**) PW.

**Figure 12 molecules-29-05953-f012:**
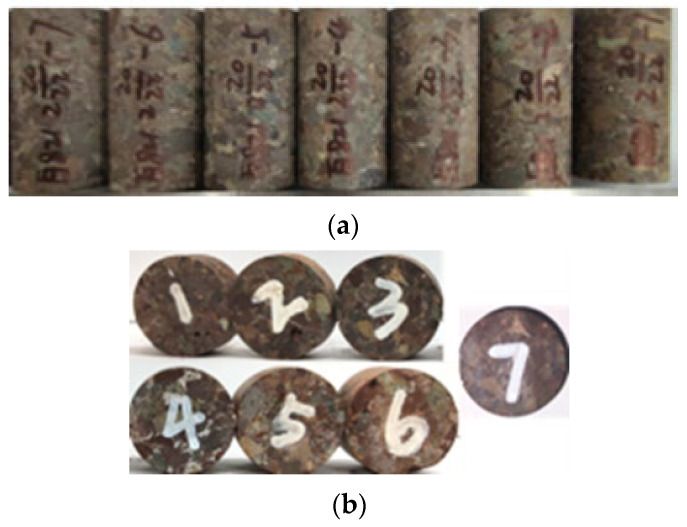
Core samples prepared for experiments: (**a**) cylindrical core samples for imbibition experiments and (**b**) sliced core samples for wettability evaluation experiments.

**Figure 13 molecules-29-05953-f013:**
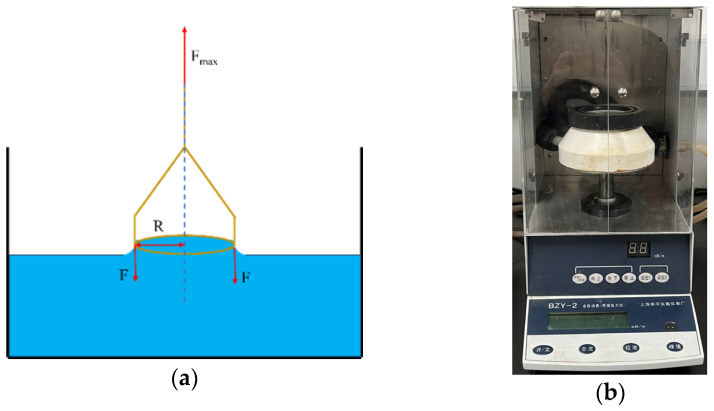
Measurements of oil–water interfacial tension: (**a**) du Noüy method; (**b**) BZY-2 micro-controlled automatic surface and interface tensiometer.

**Figure 14 molecules-29-05953-f014:**
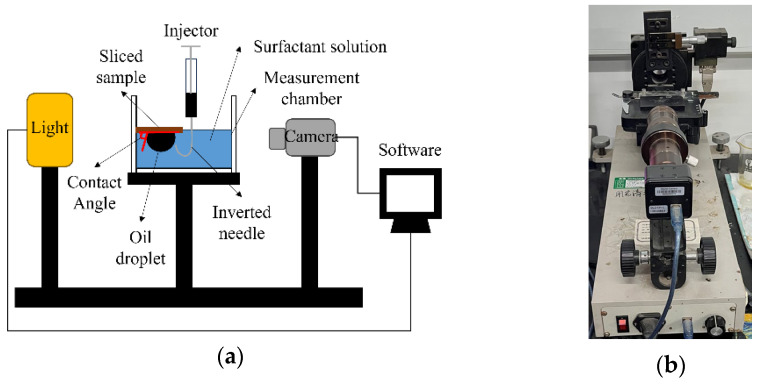
Contact angle measurements: (**a**) scheme diagram of the contact angle method; (**b**) JY-PHb contact angle measuring instrument.

**Figure 15 molecules-29-05953-f015:**
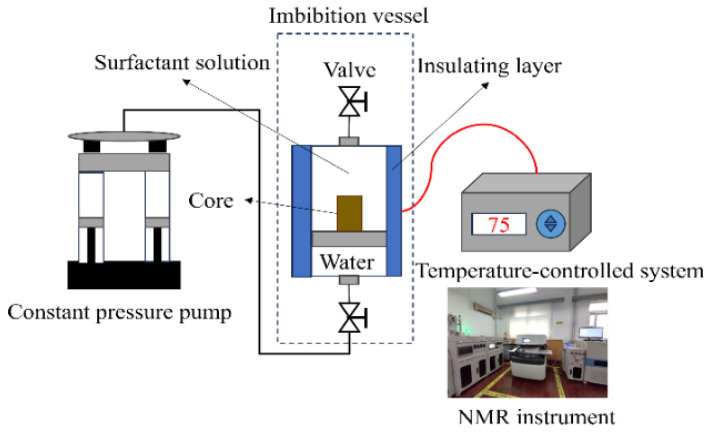
Schematic diagram of NMR-monitored HTHP imbibition device.

**Table 1 molecules-29-05953-t001:** Mineral content in the rock samples.

Mineral	Quartz	Potassium Feldspar	Plagioclase	Calcite	Hematite	Analcite	Clay Minerals
Content (%)	31.6	1.8	29.7	4.0	7.5	4.2	21.2

**Table 2 molecules-29-05953-t002:** Pore size distribution and pore classification of glutenite cores.

Pore Radius /µm	*T_2_* Relaxation Time/ms	Pore Type
0–0.01	0.01 < *T_2_* ≤ 0.69124	Micropores
0.01–0.1	0.69124 < *T_2_* ≤ 10.62293	Mesopores
>0.1	10.62293 < *T_2_*	Macropores

**Table 3 molecules-29-05953-t003:** Compositions of the produced water from the glutenite reservoir.

Total Salinity (mg/L)	Components (mg/L)							pH
	K^+^	Na^+^	Ca^2+^	Mg^2+^	Cl^−^	SO_4_^2−^	HCO_3_^−^	
10,682.45	67.00	6684	134	20.53	9178	350	2083	7.84

**Table 4 molecules-29-05953-t004:** Properties of synthetic oil and produced water at 75 °C.

Substance	Viscosity (mPa·s)	Density (g/cm^3^)
Produced water	0.55	1.09
Synthetic oil	1.15	0.88

**Table 5 molecules-29-05953-t005:** Descriptions of different types of surfactants used in the experiments.

Surfactant Type	Surfactant Name	Abbreviation	Molecular Formula	Purity (%)
Anionic surfactant	Sodium dodecyl sulfate	SDS	C_12_H_25_SO_3_Na	99
	Sodium dodecyl benzene sulfonate	SDBS	C_18_H_29_SO_3_Na	95
	Oil-displacing agent	KPS	/	/
Cationic surfactant	Cetyl trimethyl ammonium bromide	CTAB	C_16_H_33_(CH_3_)_3_NBr	99
Nonionic surfactant	Dodecane phenol polyoxyethylene ether	OP-10	C_34_H_62_O_11_	99
Nanoemulsion	/	CN-A	/	/

**Table 6 molecules-29-05953-t006:** Detailed petrophysical properties of core samples for imbibition experiments.

Sample Number	Length (cm)	Diameter (cm)	Rock Density (g/cm^3^)	Porosity (%)	Permeability (mD)	Surfactant Solutions
I1	4.99	2.52	2.53	7.19	0.0055	0.1% SDS solution
I2	4.97	2.52	2.52	7.83	0.0091	Produced water
I3	5.01	2.52	2.54	7.53	0.0025	0.1% CN-A solution
I4	4.98	2.52	2.54	7.52	0.0019	0.1% KPS solution
I5	5.00	2.51	2.53	7.68	0.0019	0.1% SDBS solution
I6	5.00	2.52	2.54	7.45	0.0022	0.1% CTAB solution
I7	4.98	2.52	2.56	7.62	0.0031	0.1% OP-10 solution

## Data Availability

The data presented in this study are available on request from the corresponding author.

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
