# Peer review of "Experimental Study on the Mechanism of Enhanced Imbibition with Different Types of Surfactants in Low-Permeability Glutenite Reservoirs"

_molecules, 2024, doi:10.3390/molecules29245953_

Round 1

Reviewer 1 Report

Comments and Suggestions for Authors

Surfactants perform well in imbibition improvement and EOR. In this study, the performance of non-ionic, anionic, and cationic surfactants and nanofluids in imbibition efficiency enhancement and imbibition recovery improvement in low-permeability glutenite reservoirs were investigated and compared. The mechanisms of EOR for different surfactants were clarified. It provides an in-depth analysis of the mechanism of surfactant-enhanced imbibition in low-permeability glutenite reservoirs, which is of great significance for the selection of suitable surfactants. This manuscript is well written, but there are some problems that need to be addressed before publication.

1. The conclusions in this study were based on core samples from glutenite reservoirs in the Junggar Basin. Is it a representative type of reservoir? Can the conclusions be applied to other reservoirs? If not, what is the difference?

2. The concentration of surfactants has effects on the EOR. Three concentrations of surfactants were configured in this study. However, why was 0.1% surfactant concentration finally chosen for the experiments?

3. Several types of surfactants were selected to compare their performance, whereas the self-developed nanoemulsion seems not very effective in improving oil recovery. Why was the nanoemulsion chosen as a comparative?

4. The EOR mechanisms of surfactants involves low interfacial tension, wettability change, and emulsification. Among all the factors, which is the controlling factor and may have the greatest influence on imbibition?

5. In the manuscript, lines 417-421, Page 13, “Specifically, the magnitudes decrease dramatically in the micropore (short relaxation time) while those in the mesopore and macropores (long relaxation time) change slightly. This demonstrates that oil was mainly distributed in the micropore initially and displaced to the mesopore and macropores by the surfactant solutions and PW and extracted from the fractures during the imbibition.”, Why is the oil in micropores mostly extracted and what is the difference among different surfactants in terms of displacing oil from micropores to mesopores and macropores?

6. What is the main innovation of this study and how was it achieved?

Author Response

Surfactants perform well in imbibition improvement and EOR. In this study, the performance of non-ionic, anionic, and cationic surfactants and nanofluids in imbibition efficiency enhancement and imbibition recovery improvement in low-permeability glutenite reservoirs were investigated and compared. The mechanisms of EOR for different surfactants were clarified. It provides an in-depth analysis of the mechanism of surfactant-enhanced imbibition in low-permeability glutenite reservoirs, which is of great significance for the selection of suitable surfactants. This manuscript is well written, but there are some problems that need to be addressed before publication.

Comment 1:

The conclusions in this study were based on core samples from glutenite reservoirs in the Junggar Basin. Is it a representative type of reservoir? Can the conclusions be applied to other reservoirs? If not, what is the difference?

Response 1:

    The Mahu conglomerate reservoir is the largest super-large tight glutenite reservoir in the world with a reserve of 10 × 108 t. As a typical unconventional glutenite reservoir, the formation properties such as rock composition, salinity, temperature and pressure are different from other types of reservoirs. The results obtained in this study are based on the experiments of imbibition and oil recovery using the samples from the glutenite reservoirs, and provide a way to enhance imbibition and oil recovery through selection of proper surfactants. Specific experiments need to be conducted and analysis are required for different types of reservoirs. This paper provides the way for the selection of surfactants. Surfactants can be optimized for different reservoirs under the same way.

Comment 2:

The concentration of surfactants has effects on the EOR. Three concentrations of surfactants were configured in this study. However, why was 0.1% surfactant concentration finally chosen for the experiments?

Response 2:

The surfactant concentration is determined by the degree of reduction of interfacial tension (IFT). Compared with produced water (PW), the addition of surfactants can greatly reduce the IFT of the solution and enhance the imbibition. As the surfactant concentration increases, the IFT of most surfactant solutions decrease (except the nanoemulsion CN-A, which has the smallest IFT at concentration of 0.1%). The IFT can be reduced to similar values at three concentrations for most of the surfactants, while for surfactants SDBS, CTAB, and OP-10, the decrease of IFT gradually slows down with the further increase of surfactant concentrations from 0.1% to 0.15%. Considering the cost of surfactants, the concentration of 0.1% was finally selected for the following experiments. We supplemented the relevant contents and marked them in blue in the revised manuscript. The corresponding contents are as follows:

Compared with produced water (PW), the addition of surfactants can greatly reduce the IFT of the solution and enhance the imbibition. As the surfactant concentration increases, the IFT of most surfactant solutions decreases except for the nanoemulsion CN-A. The IFT of CN-A is the lowest at a concentration of 0.1%. For most surfactants, IFT can be reduced to similar values at the three concentrations, while for surfactants SDBS, CTAB, and OP-10, the decrease of IFT gradually slows down with the further increase of surfactant concentrations from 0.1% to 0.15%. Considering the cost of surfactants, the concentration of 0.1% was finally selected for the following experiments. (Lines 334-341 of the revised manuscript

Comment 3:

Several types of surfactants were selected to compare their performance, whereas the self-developed nanoemulsion seems not very effective in improving oil recovery. Why was the nanoemulsion chosen as a comparative?

Response 3:

According to the experimental results in this study, the nanoemulsion may not be suitable for the glutenite reservoirs due to the poor performance in enhancing imbibition. However, nanoemulsion has been proved effective in other sandstone reservoirs because of the small particle size. It can enter, diffuse and migrate in most of the pore throat in tight reservoirs, extending the imbibition range and improve the recovery rate in micropores. In addition, it can emulsify crude oil, weaken the Jamin effect when oil drops pass through the pore throat, and reduce the seepage resistance. Moreover, it also changes the wettability for oil-wet reservoirs. Therefore, nanoemulsion was chosen as one of the options in this study. However, the stability of nanoemulsions can be affected by reservoir temperature, salinity, polymer concentration and other factors and the suitability in the glutenite reservoirs need to be studied. (Yuan et al, 2024; Sun et al, 2024; Xu et al, 2023).

Yuan, S., Zhou, F.J., Li, Y., et al. Mechanism of imbibition and production enhancement of nanoemulsion in tight sandstone oil reservoirs [J]. Petroleum Geology and Recovery Efficiency,2024,31(1):126-136.

Sun, Z.H., Li, M.H., Yuan, S., et al. The flooding mechanism and oil recovery of nanoemulsion on the fractured/non-fractured tight sandstone based on online LF-NMR experiments, Energy, Volume 291, 2024, 130226, ISSN 0360-5442, https://doi.org/10.1016/j.energy.2023.130226.

Xu, H., Li, Y., Zhou, F.J., et al. Adsorption characteristics, isotherm, kinetics, and diffusion of nanoemulsion in tight sandstone reservoir, Chemical Engineering Journal, Volume 470, 2023, 144070, ISSN 1385-8947, https://doi.org/10.1016/j.cej.2023.144070.

Comment 4:

The EOR mechanisms of surfactants involves low interfacial tension, wettability change, and emulsification. Among all the factors, which is the controlling factor and may have the greatest influence on imbibition?

Response 4:

Reducing IFT, improving wettability, and emulsifying crude oil are the most important mechanism of surfactants for EOR, whereas different surfactants perform differently in each mechanism. In this study, the imbibition recovery rate of surfactant KPS is the highest, followed by SDBS and SDS. Though the ability of OP-10 in reducing IFT is not very good, the recovery rate is second to KPS. Therefore,there is no dominant factor and all the mechanism work together to enhance imbibition and EOR. Therefore, it is necessary to consider various factors in the selection of surfactants for a specific reservoir.

Comment 5:

In the manuscript, lines 417-421, Page 13, “Specifically, the magnitudes decrease dramatically in the micropore (short relaxation time) while those in the mesopore and macropores (long relaxation time) change slightly. This demonstrates that oil was mainly distributed in the micropore initially and displaced to the mesopore and macropores by the surfactant solutions and PW and extracted from the fractures during the imbibition.”, Why is the oil in micropores mostly extracted and what is the difference among different surfactants in terms of displacing oil from micropores to mesopores and macropores?

Response 5:

    According to the T2 spectrum of the saturated core sample, micropores account for a large proportion in the glutenite reservoirs. Enhanced oil recovery in this type of reservoir is mainly to utilize the oil in the micropores. The capillary force in micropores is larger during the imbibition process. By changing the rock wettability to more hydrophilic through the addition of surfactants, imbibition in the micropores is enhanced. The oil in the micropores is moved to the large pores and produced through the fractures. However, since the capillary force is relatively small in macropores, some of the oil cannot be produced easily.

Comment 6:

What is the main innovation of this study and how was it achieved?

Response 6:

The Mahu conglomerate reservoir has large reserves and abundant resources, while the oil production is low and enhancing oil recovery is the urgent issue. Surfactants performs well on enhancing imbibition while the mechanism is not very clear for different types of surfactants. Proper surfactant need to be selected for the conglomerate reservoir. The surfactants suitable for tight glutenite reservoirs were selected through a series of experiments. The performance of each surfactant in reducing IFT, wettability alteration and oil emulsification was evaluated and its mechanism of enhancing oil recovery was clarified. This study provides a perspective in the effective development of conglomerate reservoir and a way to optimize the imbibition through addition of surfactants.

Reviewer 2 Report

Comments and Suggestions for Authors

In this work, the author investigated how different surfactants and their physical properties affect the oil recovery rate for glutenite reservoirs. Overall the article is well written and organized. I would recommend its publication with minor changes and some clarifications.

KPS and OP-10 are top surfactants to recover oil in glutenite reservoirs. The authors concluded appropriate interfacial tensions are the key drivers to high oil recovery while OP-10 and KPS have drastically different IFT. SDBS, despite having similar IFT as KPS, has much lower oil recovery rate. Can the authors elaborate more on this? Would this be associated with critical micelle concentration? If so, I would recommend measuring CMC for all surfactants as well.

Author Response

In this work, the author investigated how different surfactants and their physical properties affect the oil recovery rate for glutenite reservoirs. Overall the article is well written and organized. I would recommend its publication with minor changes and some clarifications.

KPS and OP-10 are top surfactants to recover oil in glutenite reservoirs. The authors concluded appropriate interfacial tensions are the key drivers to high oil recovery while OP-10 and KPS have drastically different IFT. SDBS, despite having similar IFT as KPS, has much lower oil recovery rate. Can the authors elaborate more on this? Would this be associated with critical micelle concentration? If so, I would recommend measuring CMC for all surfactants as well.

Response:

Many thanks for the reviewer's suggestions. According to the conclusion of this experiment, there is no obvious relationship between IFT and recovery efficiency. Reducing IFT is conducive to improving the flow of crude oil and enhancing the flow ability of crude oil in the reservoir. However, super low IFT reduces the capillary force required in the imbibition process, which is not conducive to the imbibition oil recovery process. Therefore, the reduction of IFT should be a suitable value, which should not be too large or too small. The positive effects on imbibition due to the higher IFT value and good oil emulsification capacity may be the reason that OP-10 performs better than the surfactant SDBS. The recovery of SDBS is lower than that of KPS may result from its poor emulsification performance compared with KPS. It is necessary to consider various stimulation mechanisms of surfactants comprehensively to determine the recovery efficiency. The IFT is not the sole indicative of the ability to enhance oil recovery. In addition, the reviewer has given good suggestions in terms of CMC, providing us with more perspective for the future research. The CMC for all the surfactants will be measured in the subsequent experiments.

Reviewer 3 Report

Comments and Suggestions for Authors

The article presents an overview of the complex physical properties of low-permeability glutenite reservoirs. Currently, there are no universal methods for developing heavy raw materials. As  is known,  the oil recovery rate with conventional development is low for them. Various surfactants are actively involved for enhanced oil recovery. The article examines their basic properties that lead to wettability alteration and interfacial tension reduction, changing imbibition efficiencies.

The advantages of the article include:

- a detailed analysis of the current level of development of the process with the allocation of bottlenecks requiring improvement;

- the experiments conducted are described in sufficient detail;

- control tests were carried out (from 3 to 5 times), which confirms the adequacy of the results obtained;

- the discussions provide not only general descriptions of the curves, but also specific numerical values, but most importantly, it shows how certain surfactants will subsequently affect industrial operating conditions;

- the conclusions briefly reflect the key recommendations for conducting the process.

To minor comments: I would like to recommend improving the quality of the presented figures, increasing the font of the inscriptions

Author Response

The article presents an overview of the complex physical properties of low-permeability glutenite reservoirs. Currently, there are no universal methods for developing heavy raw materials. As is known, the oil recovery rate with conventional development is low for them. Various surfactants are actively involved for enhanced oil recovery. The article examines their basic properties that lead to wettability alteration and interfacial tension reduction, changing imbibition efficiencies.

The advantages of the article include:

- a detailed analysis of the current level of development of the process with the allocation of bottlenecks requiring improvement;

- the experiments conducted are described in sufficient detail;

- control tests were carried out (from 3 to 5 times), which confirms the adequacy of the results obtained;

- the discussions provide not only general descriptions of the curves, but also specific numerical values, but most importantly, it shows how certain surfactants will subsequently affect industrial operating conditions;

- the conclusions briefly reflect the key recommendations for conducting the process.

To minor comments: I would like to recommend improving the quality of the presented figures, increasing the font of the inscriptions

Response:

We have modified the font size of figures 6-7 and 9-15 according to the reviewers' suggestions.

Reviewer 4 Report

Comments and Suggestions for Authors

The manuscript entitled «Experimental study on the mechanism of enhanced imbibition with different types of surfactants in low-permeability glutenite reservoirs» by Qu et al. is devoted to the high-temperature and high-pressure imbibition experiments in case of the tight glutenite reservoir in the Junggar Basin. The influence of three commercial but different surfactant types on the wetting properties, imbibition process and oil recovery factor were thoroughly investigated. In addition, the authors tried to discuss some aspects of the mechanism of enhanced imbibition under surfactant impact. In general, the manuscript is well-structured, clear and the most obtained results are well interpreted. The topic is relevant and the obtained results can be attractive to the wide audience of the petroleum industry. However, there are some concerns and recommendations to better the quality of the given manuscript:

1.     «Glutenite reservoir formations» must be shortly defined and the main challenges have to be addressed in introduction part

2.     The reason of diluting native oil produced from the glutenite formation with kerosene is not clear.

3.     The stability of the prepared surfactant solutions to the temperature and salinity of the PW with different concentration have to be addressed in subsection of 2.1.2. i.e., Krafft Temperature (KP) and Cloud point (CP).

4.     The contact angle measurements were carried out under reservoir pressure and temperature conditions?

5.     Figure -6: for the reliability of the obtained IFT results, the accuracy of the applied method through the % error and repeatability must be mentioned in experimental section. In addition, the reasons of low efficiency of OP-10, as well as the most effective surfactants on IFT must be proposed and commented. The obtained results are not discussed at all.

6.       Figure 7: it is better to remove the bubble illustrations, as they are not readable and the angle values are probably the average of three measurements as per described method. Were left hand angles were always equal to the right-hand angles? The obtained results can be corroborated with the results of IFT measurements.   

7.     Lines 362-364: it is better to avoid uninformative statements

8.     Should imbibition process somehow depend from the CA, IFT and emulsification? If yes, then why certain surfactants show different efficiencies during imbibition process. The issue is suggested to reflect in discussion section.   

The above remarks do not affect the overall impression from the provided study, which is performed at a high scientific level. Therefore, the manuscript is recommended for publication after minor revision.

Author Response

The manuscript entitled «Experimental study on the mechanism of enhanced imbibition with different types of surfactants in low-permeability glutenite reservoirs» by Qu et al. is devoted to the high-temperature and high-pressure imbibition experiments in case of the tight glutenite reservoir in the Junggar Basin. The influence of three commercial but different surfactant types on the wetting properties, imbibition process and oil recovery factor were thoroughly investigated. In addition, the authors tried to discuss some aspects of the mechanism of enhanced imbibition under surfactant impact. In general, the manuscript is well-structured, clear and the most obtained results are well interpreted. The topic is relevant and the obtained results can be attractive to the wide audience of the petroleum industry. However, there are some concerns and recommendations to better the quality of the given manuscript:

Comment 1:

«Glutenite reservoir formations» must be shortly defined and the main challenges have to be addressed in introduction part.

Response 1:

According to the reviewer's suggestions, we supplemented the relevant contents and marked them in blue in the revised manuscript. The corresponding contents are as follows:

Glutenite reservoirs mainly refer to the reservoirs composed of coarse clastic rock, consisting of sandstone particles with different particle size gradations (Tang et al, 2024; Qi et al, 2024). They are characterized with low porosity, ultra-low permeability, and strong heterogeneity, resulting in rapid decline in production and low oil recovery efficiency. Therefore, technologies such as reservoir stimulation, imbibition with surfactants, and other measures need to be adopted to enhance recovery efficiency. (Lines 96-102 of the revised manuscript

Tang, Y., Yuan, Y.F., Li, H., et al. Glutenite reservoir characteristics model of Permian Upper Wuerhe Formation and development in Fukang Sag, Junggar Basin [J]. Petroleum Geology & Experiment, 2024, 46(5): 965-978. D01:10.11781/ysydz202405965.

Qi, H.Y., Li, X.S., Wang, Z.Z., et al. Thick glutenite reservoir evaluation and strong heterogeneous quality control reservoir characteristics of the first member of Badaowan Formation in Maxi Slope, Jungar Basin [Jl. Science Technology and Engineering, 2024, 24(23) 9718-9728. DOI:10. 12404 / j. issn. 1671-1815. 2306010.

Comment 2:

The reason of diluting native oil produced from the glutenite formation with kerosene is not clear.

Response 2:

   After the oil is produced from the formation, gas are released from the oil, increasing the density and viscosity of the oil. Diluting native oil produced from the glutenite formation with kerosene is to keep the density and viscosity of synthetic oil consistent with the crude oil in formation conditions. According to the reviewer's suggestions, we supplemented the relevant contents and marked them in blue in the revised manuscript. The corresponding contents are as follows:

    Since gases are released from the oil after oil is produced from the formation, the density and viscosity of the oil increase. To keep the density and viscosity consistent with the crude oil in formation conditions, the crude oil produced from the tight glutenite reservoir in the Junggar Basin is diluted with kerosene with a volume ratio of 1: 3, as shown in Figure 1 (a). (Lines 120-124 of the revised manuscript

Comment 3:

The stability of the prepared surfactant solutions to the temperature and salinity of the PW with different concentration have to be addressed in subsection of 2.1.2. i.e., Krafft Temperature (KP) and Cloud point (CP).

Response 3:

We agree with the reviewer that the stability of surfactant solutions to the temperature and salinity of the PW is very important. Both Krafft Temperature (KP) and Cloud point (CP) reflect the influence of temperature on the solubility of surfactants. For ionic surfactants, the solubility increases sharply at temperatures higher than KP. For the non-ionic surfactant, when the temperature is higher than the CP, the dissolution of the non-ionic surfactant changes from completely dissolved to partially dissolved, and the solution become turbid. The KP and CP of typical commercial surfactants were addressed according to the reviewer's suggestion.

The Krafft temperatures (KP) of the ionic surfactants SDS, SDBS, and CTAB were 13-18℃, 27.6℃, and 24-27℃ respectively (Giongo and Bales, 2003; Šegota et al, 2006) and the Cloud point (CP) of the non-ionic surfactant OP-10 is 61-67℃. Since the experimental conditions are much higher than the KP of the three ionic surfactants, they can be completely dissolved in PW under the experimental temperature and this ensures the stability of these surfactant solutions. For the non-ionic surfactant OP-10, the experimental temperature is slightly higher than its turbidity point, but according to the results of this study, the stability of OP-10 is good and has a higher effect on improving oil recovery. (Lines 147-155 of the revised manuscript

C.V. Giongo, B.L. Bales, Estimate of the ionization degree of ionic micelles based on Krafft temperature measurements, J. Phys. Chem. B 107 (2003) 5398–5403, https://doi.org/10.1021/jp0270957.

Suzana Šegota, Stanka Heimer, Đurđica Težak, New catanionic mixtures of dodecyldimethylammonium bromide/sodium dodecylbenzenesulphonate/water: I. Surface properties of dispersed particles, Colloids and Surfaces A: Physicochemical and Engineering Aspects, Volume 274, Issues 1–3, 2006, Pages 91-99, ISSN 0927-7757, https://doi.org/10.1016/j.colsurfa.2005.08.051.

Comment 4:

The contact angle measurements were carried out under reservoir pressure and temperature conditions?

Response 4:

Thank you for the reviewer’s remind and suggestions. For our experiments, before the initial contact angle was measured, the rock sample was soaked in the formation water under the reservoir temperature and pressure over 36h. For the contact angel change, the rock sample was soaked in the surfactant solutions under the reservoir temperature and pressure over 72h. In addition, the wettability of all the surfactants were obtained under the same experimental conditions, and therefore the results are comparable.

Comment 5:

Figure -6: for the reliability of the obtained IFT results, the accuracy of the applied method through the % error and repeatability must be mentioned in experimental section. In addition, the reasons of low efficiency of OP-10, as well as the most effective surfactants on IFT must be proposed and commented. The obtained results are not discussed at all.

Response 5:

For the reliability of the obtained IFT results, the IFT was tested three times and averaged in value for each surfactant and PW. According to the experimental results, though the performance of surfactant OP-10 is not as good as other surfactants in reducing interfacial tension, the reduction in the interfacial tension is still considerable compared with produced water and the IFT is low enough. In addition, low IFT is not positively related to high oil recovery. The surfactant OP-10 has obvious advantages in improving oil recovery. The reason that OP-10 does not perform as well as other  surfactants in IFT reduction still needs further study. According to the reviewer's suggestions, we supplemented the relevant content and marked it with them in blue in the revised manuscript. The corresponding contents are as follows:

The measurements were conducted three times for each surfactant and PW and the value of IFT was averaged to ensure the reliability of the obtained IFT results. (Lines 203-205 of the revised manuscript

It can be seen that surfactants SDBS and KPS have good advantages in reducing IFT, followed by CTAB and SDS. Surfactants CN-A and OP-10 do not perform as well as other surfactants while the IFTs are low enough compared to PW. In terms of enhanced oil recovery, there is no obvious relationship between IFT and imbibition recovery, and other factors need to be taken into account, which will be discussed in the following sections. (Lines 328-333 of the revised manuscript

Comment 6:

Figure 7: it is better to remove the bubble illustrations, as they are not readable and the angle values are probably the average of three measurements as per described method. Were left hand angles were always equal to the right-hand angles? The obtained results can be corroborated with the results of IFT measurements. 

Response 6:

Thanks to the reviewer's suggestion. We agree that removing the bubble illustration can improve the readability of Fig.7, whereas the purpose of inserting the illustration of contact angle here is to display the measurement results and changes of CA more intuitively. The left and right contact angles are not always equal to each other in the measurement process, and the contact angle values are the average of three measurements as per described method. The obtained CA results can be corroborated with the results of IFT measurements. The reduction of interfacial tension leads to the decrease in the contact angle, improving the wettability while the CA is still affected by other factors. In this study, the effects of different surfactants on reducing interfacial tension and improving wettability are different. Specifically, the cationic surfactants SDBS and SDS perform well in reducing IFT and changing wettability, and the change value of CA is also large, followed by KPS. However, the ability of non-ionic surfactant OP-10 to reduce IFT is different from other surfactants, while the change of CA is similar to that of KPS. In addition, CTAB can greatly reduce IFT, but its ability to reduce the wetting Angle is poor. The specific correlation needs further experimental research.

Comment 7:

Lines 362-364: it is better to avoid uninformative statements.

Response 7:

    According to the reviewer’s suggestions. The uninformative statements in line 362-364 were removed from the manuscript.

Comment 8:

Should imbibition process somehow depend from the CA, IFT and emulsification? If yes, then why certain surfactants show different efficiencies during imbibition process. The issue is suggested to reflect in discussion section.  

Response 8:

Surfactants affect the imbibition through reducing ITF, changing wettability and emulsifying crude oil, which play important roles in enhancing imbibition. However, the ability of reducing ITF, changing wettability and emulsifying crude oil for different surfactants are different due to their different composition, structure and mechanism of enhancing imbibition. Therefore, they perform differently in enhancing imbibition and oil recovery. According to the reviewer's suggestions, we supplemented the relevant content and marked them in blue in the revised manuscript. The corresponding contents are as follows:

Surfactants enhance imbibition and oil recovery by reducing IFT, altering wettability, and emulsifying crude oil. Moderate reduction of IFT reduces the adhesion of oil to the rock surface, facilitating the falling off of oil droplets from the rock surface, whereas certain capillary force is still required for driving oil in oil recovery. In addition, the wettability can be altered to be more hydrophilic in favor of oil recovery and crude oil emulsification improves oil migration in the pore throat with surfactants. However, the ability to reduce IFT, change wettability, and emulsify crude oil for different surfactants are different due to their different composition, structure, and mechanism of enhancing imbibition. Therefore, surfactants perform differently in enhancing imbibition and oil recovery. It is necessary to consider all the factors for selecting surfactants to improve the oil recovery in the glutenite reservoirs. (Lines 581-591 of the revised manuscript)